# Widespread global disparities between modelled and observed mid-depth ocean currents

Fenzhen Su [1,2,3,4,14] ✉, Rong Fan[1,2,14], Fengqin Yan[1,2], Michael Meadows [3,5,6], Vincent Lyne [7], Po Hu[8], Xiangzhou Song [9], Tianyu Zhang[10], Zenghong Liu[11], Chenghu Zhou [1,2], Tao Pei [1,2], Xiaomei Yang[1,2], Yunyan Du[1,2], Zexun Wei [12], Fan Wang [8], Yiquan Qi[9] & Fei Chai[11,13]

The mid-depth ocean circulation is critically linked to actual changes in the long-term global climate system. However, in the past few decades, predictions based on ocean circulation models highlight the lack of data, knowledge, and long-term implications in climate change assessment. Here, using 842,421 observations produced by Argo floats from 2001-2020, and Lagrangian simulations, we show that only 3.8% of the mid-depth oceans, including part of the equatorial Pacific Ocean and the Antarctic Circumpolar Current, can be regarded as accurately modelled, while other regions exhibit significant underestimations in mean current velocity. Knowledge of ocean circulation is generally more complete in the low-latitude oceans but is especially poor in high latitude regions. Accordingly, we propose improvements in forecasting, model representation of stochasticity, and enhancement of observations of ocean currents. The study demonstrates that knowledge and model representations of global circulation are substantially compromised by inaccuracies of significant magnitude and direction, with important implications for modelled predictions of currents, temperature, carbon dioxide sequestration, and sea-level rise trends.

The oceans, covering more than 70% of the earth's surface and containing 95% of all water, critically modulate climate change through carbon dioxide sequestration and by absorbing excess heat[1–11]. Dynamically, the large-scale ocean circulation plays a key role in the Earth's climate system by redistributing the heat of the ocean[12,13]. Stratification arises through the processes of vertical mixing, one of the products of which is the mid-depth ocean near 1000 m between variable waters near the surface and stable waters at greater depth[14]. Interactions

[1]State Key Laboratory of Resources and Environmental Information System, Institute of Geographic Sciences and Natural Resources Research, Chinese Academy of Sciences, Beijing 100101, China. [2]College of Resources and Environment, University of Chinese Academy of Sciences, Beijing 100049, China. [3]School of Geography and Ocean Sciences, Nanjing University, Nanjing 210093, China. [4]Collaborative Innovation Center for the South China Sea Studies, Nanjing University, Nanjing 210023, China. [5]Department of Environmental & Geographical Science, University of Cape Town, Rondebosch 7701, South Africa. [6]College of Geography and Environmental Sciences, Zhejiang Normal University, Jinhua 321004, China. [7]IMAS-Hobart, University of Tasmania, Tasmania 7004, Australia. [8]Key Laboratory of Ocean Circulation and Waves, Institute of Oceanology, Chinese Academy of Sciences, Qingdao 266071, China. [9]College of Oceanography, Hohai University, Nanjing 211100, China. [10]College of Oceanography and Meteorology, Guangdong Ocean University, Zhanjiang 524088, China. [11]State Key Laboratory of Satellite Ocean Environment Dynamics, Second Institute of Oceanography, Ministry of Natural Resources, Hangzhou 310012, China. [12]First Institute of Oceanography, and Key Laboratory of Marine Science and Numerical Modeling, Ministry of Natural Resources, Qingdao 266061, China. [13]College of Ocean and Earth Sciences, Xiamen University, Xiamen 361102, China. [14]These authors contributed equally: Fenzhen Su, Rong Fan. ✉e-mail: sufz@lreis.ac.cn

between surface currents and thermohaline currents inject changes into the mid-depth ocean near 1000 m. In turn, mid-depth ocean circulation has a significant impact on the global ocean as a whole by altering the physical characteristics of ocean waters[15–18] that play a fundamental role in the longer-term evolution of Earth's climate system. However, knowledge of mid-depth ocean circulation is hampered by the lack of direct observations and novel methods are therefore needed to improve data, modeling accuracy in order to resolve the mechanism of mid-depth ocean circulation and related processes.

In the past two decades, over 4000 floats from the Argo program provided more than two million profiles of water column properties from the upper 2000-m depth of the global ocean[10,19,20]. These enable the construction of satellite-tracked sea-surface profiles and cycles for direct estimation of mid-depth ocean velocities, and spatio-temporal circulation patterns from regional to global scales[20–33]. Argo velocities derived in this manner are Lagrangian, whereas ocean circulation models are Eulerian, resulting in systematic biases of current velocity when the coordinates are transformed[34–36]. Based on simulated Argo experiments[37] it has been suggested that Argo floats accelerate near strong mean flows due to the eddy-mean flow interaction effect, although other potential biases and their implications for models remain highly uncertain. Using comprehensive and quantitative methods, we present here the detailed assessment and validation of global ocean circulation model skills near 1000 m depth.

## Results

### Sparse well-modelled mid-depth ocean

Lagrangian velocities near 1,000 m depth for all the earth's oceans were computed and multiple accuracy indicators were used to compare Argo float velocities with simulated velocities from global circulation models. The assessment is based on 842,421 Argo float observation displacements over 20 years, together with float simulation experiments. The goal is to map the areas of the oceans that are more reliably modelled and classify them in terms of how much is known about spatial variations in mid-ocean circulation. Global 1000-m ocean velocities were obtained from three eddy-permitting and daily-mean ocean general circulation models, including ECCO2 (Estimating the Circulation and Climate of the Ocean model, Phase II), OFES (the Ocean General Circulation Model for the Earth Simulator) and CMEMS GLORYS12 (Global Ocean Reanalysis and Simulations), and Argo velocities were acquired from the ANDRO dataset. Indicators of differences (see Supplementary Materials for details) comprise: DOD (Difference of Direction); DOV (Difference of Velocity); PDOV (Percentage of Velocity Difference); SD (Separation Distance); SS (Skill Score based on the cumulative simulated separation distance normalized by the cumulative observed displacement). We describe the distribution of each of these statistics before presenting the results of cluster analysis to reveal the linkages between indicators and overall spatial patterns of the clusters.

Fig. 1 shows the global spatial distribution of the well-modelled areas, defined as areas where DOD is less than 30° and PDOV is less than 50%. The two most critical indicators are adopted, and the effect of parameter thresholds on the results can be seen in Supplementary Table 1. Overall, only 1.3–3.8% of the global oceans are considered to have a robust or reliable prediction of the 1000-m depth circulation and hence, improved velocity field magnitude and direction data are needed for more than 96% of the global ocean. This deficiency therefore calls into question the reliability of predictions of the long-term implications of mid-depth ocean circulation changes. Meanwhile, several regions are better modelled, including the Antarctic Circumpolar Current and the equatorial Pacific Ocean, which is the largest spatially continuous well-modelled area that enables more reliable estimates of ocean current dynamics in the 1000-m layer.

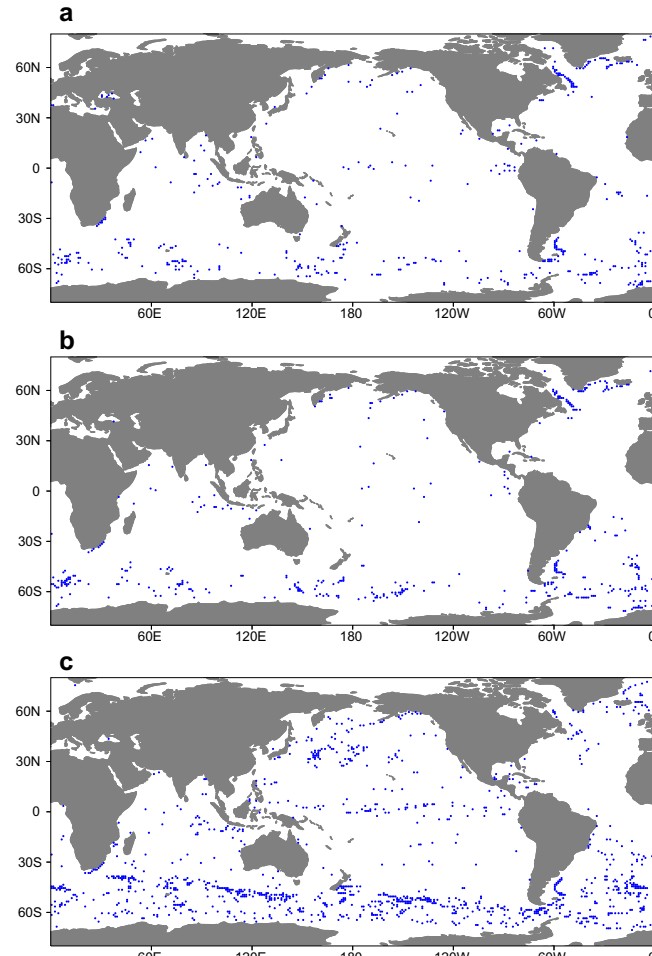

**Fig. 1 | Global distribution of well-modelled areas for models, 1.415% for ECCO2(a), 1.325% for OFES(b), and 3.843% for CMEMS(c).** The well-modelled areas are highlighted by blue dots.

### Spatial patterns of widespread global disparities

The global distribution of DOD in Fig. 2 shows that DOD in at least 81.1% of the oceans exceeds 45° or more (as shown by the most accurate model CMEMS), meaning that accurate simulation of circulation direction at 1000-m is lacking in more than four-fifths of the global oceans. However, there are considerable areas with better representation, where DOD is below 45°, notably in the equatorial Pacific Ocean and the Indian Ocean but also parts of the Antarctic Circumpolar Current, the Agulhas Current, the Gulf Stream and the Brazilian Current (as the darkest cool colors in Fig. 2 shows).

The global spatial distribution of DOV for ECCO2 and OFES in Fig. 2 and Supplementary Fig. 2 shows that observed velocities are greater than simulated values near the equator (6°N-10°S) and especially across most of the western boundary current regions. Globally, DOV for ECCO2 exceeds 3 cm/s across 51.12 million km². Regions with the most significant positive DOV anomalies include the Agulhas Current and Gulf Stream, where DOV reaches its highest values (16 cm/s) and PDOV reaches a maximum of 92.7%. In the Antarctic Circumpolar Current regions, positive and negative values of DOV are staggered, but negative PDOV dominates the overall spatial pattern and simulated velocities are in general much greater than observed values and more than twice the observed velocities across 1.02 million km² of the global oceans (PDOV > 100%). As for CMEMS (Fig. 2-f), similar underestimation of velocities is shown near the equator and parts of the western boundary current regions.

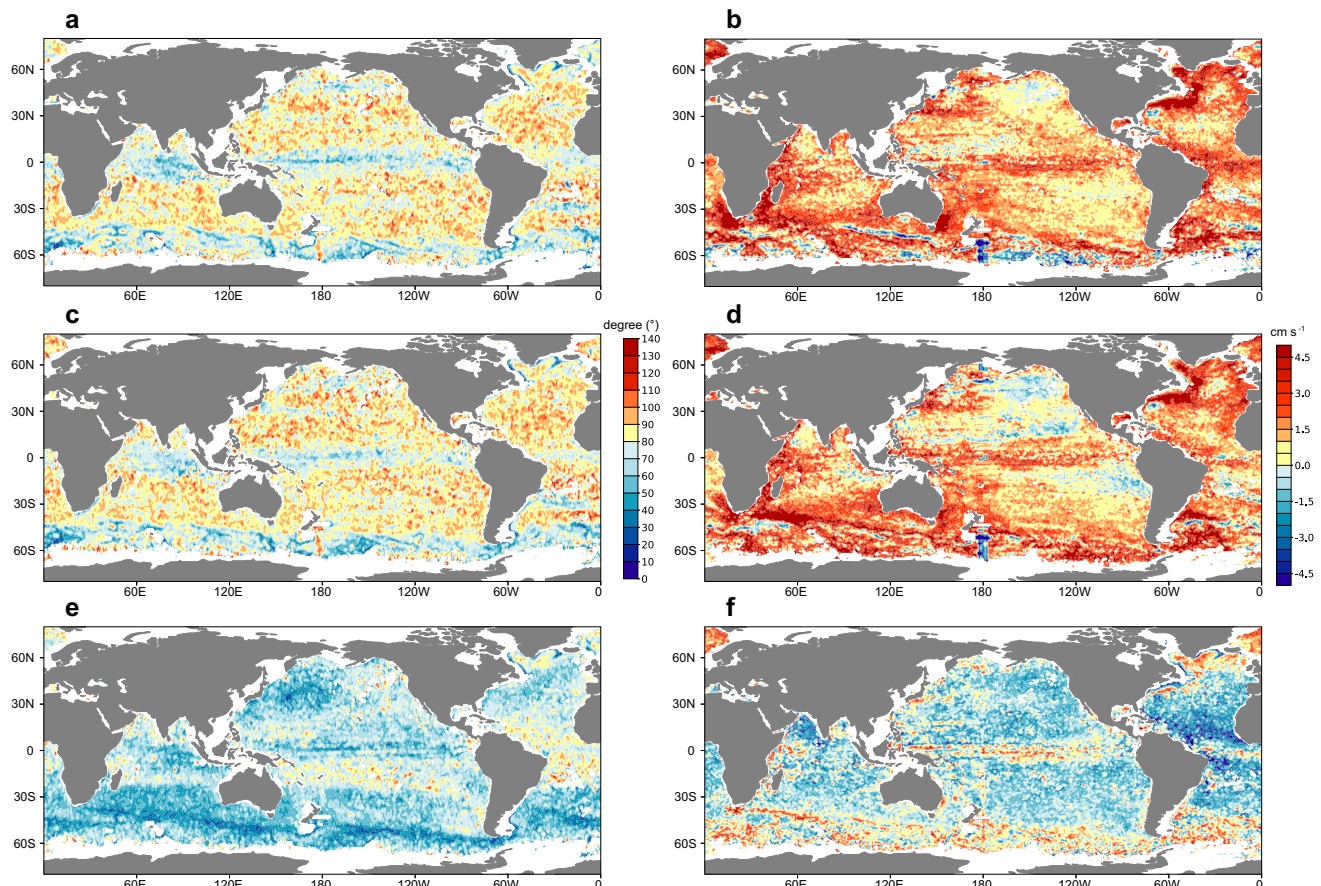

**Fig. 2 | Global spatial distribution of DOD (Difference of direction) and DOV (Difference of velocity) for ECCO2 (a, b), OFES (c, d), CMEMS (e, f).** The (a, c, e) panels: mean DOD during 2001–2020 (shaded color). Warm (Cool) color indicates that observed velocity of float displacements exceeds (lags) simulated velocity. The (b, d, f) panels: averaged DOV during 2001–2020 (shaded color).

Significant kinetic deviations exist in zonal and meridional velocity (Supplementary Fig. 10) and in the mean velocity fields (Supplementary Fig. 11). These systematic biases may be caused by lack of eddy kinetic energy (Supplementary Fig. 12). Significant underestimation of eddy kinetic energy is observed in almost all the western boundary current regions.

We also analyzed the SD and SS to reflect an overall combined accuracy statistic. As shown in Supplementary Fig. 3 and Supplementary Fig. 4, the spatial pattern of SD corresponds very well with observed velocity measurements (Supplementary Fig. 11-a), high SD occurs in the Southern Ocean, in the western boundary current regions, and near the equator. The SS in the low-latitude oceans is generally greater than in other oceans in ECCO2 and OFES, although sporadic high SS signals are also seen in the middle to high-latitude oceans. Meanwhile, SS for CMEMS shows high SS signals in both low-latitude oceans and Antarctic Circumpolar Current regions.

The results suggest that there is a relatively accurate prediction of circulation characteristics across large areas of the low-latitude oceans, *viz.* −8°N to 8°S (Fig. 1). Overall, relatively high SS suggests adequate agreement between observed and simulated current directions (DOD < 45°, Fig. 2) in most of the equatorial Pacific Ocean, the north Indian Ocean, and part of the middle Atlantic Ocean. Positive continuous spatial characteristics of SS (>0.5) are evident in both the equatorial Pacific Ocean and the northern Indian Ocean. In addition, there are positive signals (such as DOD and SS) indicating that the Antarctic Circumpolar Current regions are better modelled, but the evidence is relatively weak and even then only for the CMEMS simulation.

In the mid-latitude oceans, the strong and persistent western boundary currents show characteristics different to those of other open oceans. In these regions, both simulated and observed velocities are high (Supplementary Fig. 11). DOD results are poor (typically > 80° in ECCO2 and OFES) in most of the mid-latitude oceans. Moreover, there are markedly positive DOV signals (> 3 cm/s) in the main pathways of the Agulhas Current, Gulf Stream, Kuroshio Current, Brazilian Current and East Australian Current. For the Gulf Stream (together with its extension), observed velocities exceed those simulated in all models, even up to 70% (Supplementary Fig. 2). In addition, SS is generally low (SS < 0.4) across the mid-latitude oceans.

Of the high-latitude oceans, the Southern Ocean, most especially around the Antarctic Circumpolar Current (ACC) system, exhibits relatively good agreement between simulated and observed displacements (Fig. 1; Fig. 2). The core pathways of the ACC are reflected in the mean velocity of Argo floats, revealing strong jet currents, which are also evident in the south Indian Ocean, south Pacific Ocean and southwest Atlantic Ocean (Supplementary Fig. 11-a). Several of the indicators exhibit clear, spatially continuous patterns along the current pathways, with lower DOD, higher SD and both positive and negative DOV signals. Exceptionally high simulation velocities (PDOV is < −100% in some cases) are a key feature of these regions. But SS values differ between models, showing lower signals in ECCO2 and OFES, and higher in CMEMS (generally above 0.5).

## Statistical characteristics of widespread global disparities

Based on statistical K-means clustering analysis of the indicators (see Supplementary Table 2), the global ocean yields four clusters, each

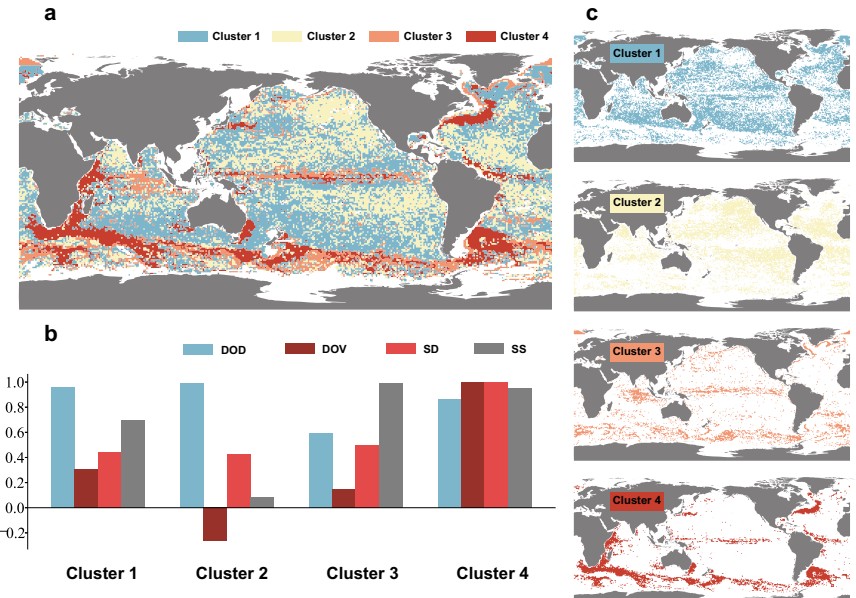

**Fig. 3 | Multivariate K-means clustering analysis results for 4 clusters. a** The clustering map shows spatial distribution, and (**b**) the column chart summarizes normalized indicators of displacement disparity for each cluster. **c** The separate clustering maps show the spatial distribution of each cluster. The clustering indicators are DOD (Difference of direction), DOV (Difference of velocity), SD (Separation distance) and SS (Skill score).

with a characteristic coherent profile (Fig. 3). A visual assessment of the clusters in Fig. 3-b leads to the following observations relating to accuracy and spatial distribution of the different indicators. Clusters are clearly distributed along latitudinal zones, although the boundary current regions, particularly evident in Cluster-4, do extend longitudinally. Spatial patterns are strongly evident, especially for Cluster-3 and Cluster-4 which are distributed in low latitudes and higher latitudes, while other cluster areas mainly occur at mid-latitudes.

The largest cluster, Cluster-1, covers 40.4% of the oceans with significantly higher DOD, and is widely distributed in the mid-latitudes, including the extended boundary currents regions. Although distributed in similar latitudes, Cluster-2 exhibits poorer ocean current accuracy, with the highest values for DOD and lowest values for SS (average below 0.2), while DOV in this cluster is negative, indicating that velocity is markedly overestimated here. Cluster-3 occupies 14.2% of the oceans, including parts of the equatorial Pacific Ocean, the north Indian Ocean and the Antarctic Circumpolar Current, where DOD, DOV and SD all lie below global mean values. Ocean current accuracy is adequate in regions where this cluster is located. Cluster-4 has the highest DOV and SD values along with higher velocity, and is located across almost all of the Western Boundary Currents and their extensions, as well as most of the Antarctic Circumpolar Current. Accordingly, the ocean current velocity in these regions is severely underestimated.

## Discussion

Overall, our integrated assessment analyses show significant and extensive discrepancies between modelled and observed velocity fields across the oceans, mainly associated with energy underestimation and direction bias. On a global scale, those areas of the oceans that are more accurately modelled include the equatorial Pacific Ocean, the northern Indian Ocean, and part of the Southern Ocean. Regionally, the low-latitude oceans including the equatorial Pacific Ocean, the north Indian Ocean and the middle of the Atlantic Ocean exhibit better agreement in terms of both amplitude and direction of circulation dynamics compared with the middle and high-latitude oceans. Estimation of circulation needs to be improved in the mid- to high-latitudes, where underestimation of both average current

energy and direction deviation is evident. The Western Boundary Currents, the most prominent features of mid-latitude ocean circulation, are known for their complex current structure and complex topographic interaction processes and prove particularly challenging to estimate accurately.

Of particular importance is the finding that circulation energy is underestimated across almost all of the global oceans. This arises in part because high-frequency dynamics are poorly resolved in ocean circulation models and also due to the inadequate solutions of sub-grid processes, such as the lack of eddy kinetic energy illustrated in Supplementary Fig. 12. Moreover, parameterizations of sub-grid processes result in a series of errors, a situation which remains still the most advanced challenge in ocean modelling[38,39]. Taking tidal processes as an example, previous studies have confirmed that tidal currents traversing rough bathymetry could induce turbulence and mixing to accelerate mid-depth ocean circulation[40–42].

Our analysis further demonstrates that circulation accuracy tends to be poor in oceans with slower mean currents, such as the north Pacific Ocean and the north Atlantic Ocean. There is evidence of intrinsic chaotic variability in these oceans that is not easily resolved by ocean circulation models[43]. Mesoscale features and unstable processes such as eddies and instability waves may also result in simulation errors especially in regions of weak amplitudes[44].

As noted by Fox-Kemper, Adcroft et al.[38], numerical and parameterization improvements will continue to define the state of the art in ocean circulation modelling. In the future, we expect ocean circulation models to represent observed quantities of ocean flows more faithfully through: (a) More intensive and qualified observations. There is a pressing need for more observations in the mid-depth ocean. A proposal to improve the sampling density of the Argo program particularly in parts of the world ocean where climate impacts are especially strong, was raised by Riser, Freeland et al.[20]. Here we recommend enhancement of observations in the mid-depth ocean, such as by increasing the float deployment and adding an inertial navigation component in float design which can provide locations at greater frequency during the parking period. (b) More productive parameterization. The ocean is a nonconstant and nonuniform dynamical system with high-frequency variations. Thus, productive

parameterization is particularly challenging and much needed. To strengthen the prediction skill of mid-depth ocean currents, the parameterization of global modelling is expected to be improved, such as depth-dependent eddy diffusivity and viscosity[45,46], internal mixing with spatio-temporal variability[47], ice-ocean basal friction[48,49]. (c) Finer resolution. With the rapid growth of high-performance computing techniques and cloud-based platforms, higher-resolution global oceanic circulation modelling systems may provide better estimates for certain regions[50–52]. (d) Last but not least, we would like to point out the recent breakthrough in modelling large-scale ocean currents through exact solutions to the geophysical fluid dynamics governing equations[53–57]. This kind of pioneering mathematical analysis of geophysical currents may prompt further in-depth theoretical studies[58].

Possible sources of uncertainty in this study are as follows. Firstly, observation of absolute velocity based on the Argo float satellite-tracked system (ARGOS and GPS) is subject to a systematic displacement error of between 8 to 1500 m[33,59]. Unpredictable float movement at the ocean surface occurs frequently due to wind, wave and surface current processes. Shear displacement caused by vertical currents during descending and ascending is an additional source of error. The ANDRO dataset has been partially corrected for both types of error and the error could be further adjusted, for example using an Argo simulation system to derive accurate systematic errors[37]. A further caveat is that ocean circulation models generate mean conditions and are not suited to resolving fine-scale spatial and high-frequency temporal variability. The output of ocean circulation models used in our work has similar deficiencies. Notwithstanding these uncertainties, our study offers important insights that improve the awareness of the representational accuracy of ocean circulation in models, and enhances regional observations.

The study highlights discrepancies between modelled and observed ocean circulation in terms of spatial structure, intensity, and direction that must be accounted for in ocean modeling and prediction. In so doing, our analysis exposes those areas of the mid-depth ocean that are most poorly resolved in terms of circulation and that should, therefore, be prioritized for observation and simulation towards the goal of optimizing ocean circulation models.

In conclusion, mean current velocities in the mid-depth oceans appear to be significantly underestimated due to the fact that high-frequency dynamical processes, such as tidal mixing, are not well captured in simulations. Although the equatorial Pacific Ocean, north Indian Ocean and part of the Southern Ocean are better modelled, excessive direction biases (over 75°) are detected across more than 60% of the global oceans, especially in areas where the circulation system is weak. Given deficiencies in the observational record, such stochasticity needs to be incorporated into ocean models through parameterization of eddy diffusivities and mixing parameters. Clear patterns of spatial clustering are revealed, with characteristics for well-modelled oceans of several assessment indicators changing rapidly with increasing latitude, on average. The major contribution of this study lies in revealing the nature and scale of the disparity between our scientific knowledge of ocean circulation and the actual ocean environment. Our analysis can help guide recommendations for more intensive observation and modeling approaches in order to decrease the extensive and significant biases between models and observations.

## Methods

### Argo displacements dataset
Calculation of 1000-m current velocities from Argo floats relies on measurement of displacement, which is the difference between the location of the last surface-fixed position before departure from the sea surface and the first fixed position after arrival at the sea surface, together with the time interval, typically 9–10 days[20,33,60,61]. For observations, in this study we use the ANDRO (Argo New Displacements Rannou and Ollitrault) displacements[59]. The ANDRO has over 843,000

displacements in individual cycles operating from September 1995 to July 2020. Maximum coverage density is 4.32 floats/100 km². ANDRO covers 86.8% of the global ocean. Of these areas, 34.7% have a total parking time of 300 days, 60.3% have 200 days, and 84.8% have 100 days.

### Global ocean circulation models velocity dataset
Global velocity fields used for Lagrangian tracking analysis are provided by the ECCO2 (Estimating the Circulation and Climate of the Ocean model, Phase II), OFES (the Ocean General Circulation Model for the Earth Simulator) and CMEMS GLORYS12 (Global Ocean Reanalysis and Simulations).

ECCO2 produces a physically consistent estimate of global ocean circulation. The circulation model assimilates a huge number of in-situ and satellite observations based on the Massachusetts Institute of Technology general circulation model (MITgcm), using Green's function optimization[62]. Here, we used the three-day current velocity field datasets from 2001 to 2020, with an eddy-permitting horizontal resolution (approximately 18 km), and 50 vertical levels within the depth range 5 to 5906 m.

OFES can perform fifty-year integrations of eddy-resolving ocean simulations in the world ocean[63]. It is based on the Modular Ocean Model (MOM3) which was developed at the Geophysical Fluid Dynamics Laboratory/the National Oceanic and Atmospheric Administration (GFDL/NOAA). The output over 2001-2019 with 1-day average and 1/10° horizontal resolution is used. OFES simulations were conducted on the Earth Simulator under the support of JAMSTEC (Japan Agency for Marine-Earth Science and Technology).

CMEMS GLORYS12 reanalysis provided a 3D description of the ocean circulation at the mesoscale[64]. It was designed and implemented using the current real-time global forecasting CMEMS (Copernicus Marine Environment Monitoring Service) system and driven by the NEMO3.1 ocean/sea-ice general circulation model. Ocean observations were assimilated by a reduced-order Kalman filter, including along-track altimeter sea-level anomaly, satellite sea-surface temperature, sea-ice concentration, and in-situ temperature and salinity (T/S) vertical profiles. The reanalysis covered the 1993-2020 period, with a 1/12° horizontal resolution and 50 vertical levels. The daily mean velocity output was used in this work.

### Regional ocean circulation model velocity dataset
Regional velocity fields used for the kinetics comparison test are provided by the DopAnV3R3-ini2007[65,66] (Doppio Analysis Version 3 Release 3). The DopAnV3R3-ini2007 is a ROMS-based (Regional Ocean Modeling System) data assimilative reanalysis focusing on the Mid-Atlantic Bight and Gulf of Maine regions of the northwestern North Atlantic. The reanalysis uses four-dimensional variational (4D-Var) data assimilation (DA) derived from comprehensive observational data from in situ platforms, coastal radars, and satellites[66]. The velocity fields output from the DopAnV3R3-ini2007 (available online https://www.seanoe.org/data/00785/89673/) is provided on a 7-km horizontal grid within 40 vertical levels from the coastal ocean to deep sea and covers the period 2007-2021.

### Kinetics comparison test
To address potential underpowering below the mesoscale of global ocean circulation models, velocity fields output from the DopAnV3R3-ini2007 (a ROMS-based reanalysis), is used to compare with the velocity fields output from global ocean circulation models. Lagrangian simulation analysis mentioned in the next section is used here. The simulation traverses the Gulf Stream zone and covers 2014–2018, viz. the 5 years that exhibit the largest Argo displacements, while no diffusivity is set here. For each model, 1102 simulated floats (equal to displacements number of displacements available to Argo) are released. The velocity comparison result between simulated floats

advected by the ROMS model and global models is shown in Supplementary Fig. 5.

## Lagrangian simulation

At the parking stage, Argo floats drift near the 1000 m depth. To reproduce the behavior of real floats during its parking stage, Lagrangian particle tracking analysis was applied. Lagrangian tracking analysis has been used in many studies to model seawater motion and material transport[67]. We used the freely available open-source Lagrangian particle trajectory tool OpenDrift[68] to compute simulated float trajectories. The spatial and temporal velocity fields were smoothed using bilinear interpolation. A second-order Runge-Kutta scheme was adopted as the advection method to derive numerical particle positions (we refer to the simulated Argo float as a "numerical particle"). The analysis tool is designed to operate in off-line mode, so computation of simulated float trajectory relies on the Eulerian velocity field output from the ocean circulation model. The simulations of different models were added with varying diffusivity. The description of the random diffusivity term is in the next section.

In our experiment, numerical particles were advected with the velocity field output of global ocean models. Each Argo cycle where parking depth lies between 950 and 1050 dbar has a corresponding numerical simulation particle. To ensure a strict comparison with real Argo floats, the uniform starting position, parking depth and parking duration of the observed trajectories were set for all numerical particles. The time step of forward numerical particle trajectory prediction was set to 1-hour and prediction results were stored at 1-day intervals. This Lagrangian simulation allows the generation of float clusters, improving the statistical significance of the outputs. A number of virtual floats is released at the starting position of each Argo cycle. The centroid of the displacement probability density cloud is considered to be representative of the float cluster displacement. The size of cluster (that is, the number of virtual floats for each release) is determined by a spatial coverage test as shown in Supplementary Fig. 8. We selected the velocity field of OFES as it has the largest diffusivity among the models, thereby ensuring the robustness of the value. Given the large number of simulations performed, we selected 100 as the size of the cluster to balance statistical accuracy and computational intensity.

## Stochastic diffusivity test

As emphasized in a series of other studies, the diffusivity of Lagrangian analysis is a significant issue[67,69,70]. In general, the stochastic term is used in Lagrangian analysis to model stochastic fluctuations, including subgrid scale diffusion and unresolved physics such as eddies, waves, turbulence or mixing processes. A map of diffusivity may be estimated by floats observations[71–73]. However, different diffusivities have been set up for ocean general circulation models[63,64]. So here horizontal isotropic diffusivities account for the dynamic processes that are not fully resolved by the global ocean circulation models[37]. The stochastic walk of the particle tracking is implemented by the Wiener noise process.

To quantify the suitable diffusivities, a stochastic diffusivity test was conducted by Lagrangian simulation only in the Gulf Stream zone. As previous studies noted, validations of high-resolution model simulations are essential to ensure the applicability of the stochastic term[67,73]. We conduct the test through comparing the Lagrangian simulation results between global models and ROMS. No diffusivity was set for ROMS simulation.

Supplementary Fig. 6 visualizes the changes of kinetics indicting by absolute velocity for each model when the diffusivity varies. For simulations of ECCO2 and OFES, their kinetic distributions gradually approach that of ROMS as the diffusivity increases, with the frequency tending to decrease for small velocities (<10 cm s⁻¹) and increase for large velocities (>10 cm s⁻¹). It seems that the applicable value is

around 1000 to 1500 m² s⁻¹. Further tests, detailed below, may provide accurate values. Instead, the kinetic distributions of CMEMS GLORYS12 simulations moves away from that of ROMS as the diffusivity leaves zero. Accordingly, the stochastic diffusivity is set to $10^{-5}$ m² s⁻¹, which is indeed close to zero[74]. It makes sense for a non-zero diffusivity which allows floats to deviate from deterministic trajectories driven by sub-grid scale diffusion.

To further test the diffusivity value for ECCO2 and OFES, correlations of probability density functions of velocities are calculated as shown in Supplementary Fig. 7. Linear correlations of each simulation of different diffusivity with the ROMS simulation are calculated. According to the maximum correlation, the diffusivity for ECCO2 is 1100 m² s⁻¹ and the value for OFES is 1400 m² s⁻¹. The diffusivities here demonstrated reasonable agreement with the global average value estimated by Cole et al.[75]. Moreover, the Kolmogorov-Smirnov test results show that the simulated kinetic distribution is consistent with that of the observed kinetics.

## Quantitative difference framework

To assess the modelling accuracy of global ocean circulation, a quantitative difference framework was used. The framework was designed for displacement comparison between observed and simulated floats.

Three kinds of indicators quantify circulation differences using float movement characteristics. The first indicator is DOD (Difference of Direction), which represents the angle between two vectors derived from observed and simulated displacement. The geometric meaning of DOD is similar to the cosine similarity used in trajectory similarity studies[76]. The DOD is calculated as follows:

$$DOD = \text{atan}\left(y_{v_{obs}}, x_{v_{obs}}\right) - \text{atan}\left(y_{v_{sim}}, x_{v_{sim}}\right) \quad (1)$$

where $v_{obs}$ and $v_{sim}$ are velocity vectors (in x, y space) exported from observed and simulated float displacement, respectively.

Secondly, velocity related indicators include DOV and PDOV. DOV (Difference of Velocity) is the magnitude of velocity difference between observed and simulated displacements:

$$DOV = |v_{obs}| - |v_{sim}| \quad (2)$$

PDOV (Percentage of Velocity Difference) represents the percentage-ratio of velocity difference compared to absolute observed velocity:

$$PDOV = \left(\frac{|v_{obs}| - |v_{sim}|}{|v_{obs}|}\right) \times 100\% \quad (3)$$

The third indicator included two universal displacement measures to quantify the movement consistency of the floats. SD (Separation Distance) is the distance between the end points of the simulated and observed float displacements, demonstrating movement differences directly. SS (Skill Score) is based on the cumulative simulated separation distance normalized by the cumulative observed displacements length[76]:

$$SS = \begin{cases} 1 - \frac{c}{n}, & (c \le n) \\ 0, & (c > n) \end{cases}, \quad (4)$$

where n is a tolerance threshold, c is a normalized cumulative Lagrangian separation distance. An estimate of c is obtained by dividing the cumulative Lagrangian separation distance (SD) by the cumulative length of the observed displacement (L):

$$c = \sum_{i=1}^{N} SD_i \bigg/ \sum_{i=1}^{N} L_i, \quad (5)$$

where $i = 1, 2,..., N$, and $N$ is the total number of days. As for the tolerance threshold ($n$), we selected $n = 1.8$, considering the expectations and requirements of the simulated model. $SS$ has been used in several other studies to assess numerical ocean circulation model accuracy[77–79]. In general, SS varies between 0 and 1, whereby higher SS values indicate improved prediction skills.

Calculations of difference are based on the match-up displacement pair dataset. With the dataset, all indicators are rasterized and adjusted to a resolution of 1° × 1°. The python-based library Cartopy was used to set map projections and map evaluation results. The multivariate K-means clustering method was adopted to explicitly extract regions with similar biases, seeing detailed statistics in the later section (several compound parameters statistics).

### Observation data investigation
The significance of difference assessments between observation and simulation relies on the number of Argo float observations in each statistical unit (1° × 1°). The spatial coverage of Argo observations with different amounts is shown in Supplementary Fig. 1.

### Several compound parameters statistics
To investigate the influence of the evaluation thresholds on the analysis, we computed compound evaluation thresholds across statistics. When DOD and DOV lie between 30–90° and 10–70%, respectively, areas of the ocean classified as well-modelled comprised 0.27-59.85% of the total (Supplementary Table 1).

Multivariate K-means clustering of normalized indicators (DOD (°), DOV (cm s-1), SD (km), SS (score)) were used to explore the linkages between indicators, and their spatial distribution. Indicators were normalized to have a mean of zero and standard deviation of one. We chose 4 clusters as increasing it to 5 produced two clusters with similar profiles and broad distributions to Cluster 1, shown in Supplementary Table 2 which also summarizes the overall profile and %Area for each Cluster.

## Data availability
All observation and model data that support the findings of this study are available as follows. The ANDRO (Argo New Displacements Rannou and Ollitrault) displacements available online (https://www.seanoe.org/data/00360/47077/) on the marine sciences data platform SEANOE (SEA scieNtific Open data Edition). The ECCO2 output is available online (https://www.ecco-group.org/). The OFES output is available online (http://apdrc.soest.hawaii.edu/erddap/griddap/hawaii_soest_b0d2_0156_7195.html). The CMEMS GLORYS12 reanalysis is delivered through the Copernicus Marine Environment Monitoring Service (CMEMS) available at https://resources.marine.copernicus.eu/product-detail/GLOBAL_MULTIYEAR_PHY_001_030. The DopAnV3R3-ini2007 velocity field output is available at https://www.seanoe.org/data/00785/89673/.

## Code availability
All analyses in this manuscript are reproducible, instructions and scripts could be found in the repository Middepth_currents_validation[80] (https://doi.org/10.5281/zenodo.7765311).

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

## Acknowledgements

This work is supported by the Strategic Priority Research Program of Chinese Academy of Sciences (Grant No. XDA19060304, F.S.) and the National Natural Science Foundation of China (Grant No. 41890854, F.S.).

## Author contributions

Conceptualization: F.S. Data curation: R.F. Funding acquisition: F.S. Investigation: F.S., F.Y. Methodology: F.S., R.F., F.Y., V.L. Visualization: F.S., R.F., V.L. Project administration: F.S., F.Y. Software: R.F. Writing, original draft: R.F., F.Y. Review and editing: F.S., R.F., F.Y., V.L., M.M., P.H., X.S., T.Z., Z.L., C.Z., T.P., X.Y., Y.D., Z.W., F.W., Y.Q., F.C. All authors subsequently reviewed the manuscript and read and approved the final version.

## Competing interests

The authors declare no competing interests.
