## [Peer Review File · Nature Communications]

Widespread global disparities between modelled and observed mid-depth ocean currentsEditorial Note: Parts of this Peer Review File have been redacted as indicated to remove third-party material where no permission to publish could be obtained.

REVIEWER COMMENTS

Reviewer #1 (Remarks to the Author):

The manuscript "Widespread global disparities between modelled and observed mid-depth ocean currents" aims at assessing the skills of global ocean currents reanalysis near 1000 dbar, by comparing the observed displacements of ARGO floats with trajectories of virtual particles integrated in the modelled velocity fields. Although the topic tackled here (the poor skills of coarse-resolution models at depth) is of importance, in my opinion, the methods used here are not appropriate and I can not recommend publication. I would however like to insist on the importance of assessing global circulation model skills, particularly in or below the thermocline, and to encourage the authors to pursue their efforts in that direction. My major objections about the methods, as well as a list of minor comments are provided below.

Major comment :

The authors compare ARGO float displacements with virtual particles integrated in modelled velocity fields for each single available float displacements. They then compute statistical properties of the angle and velocity magnitude differences. Although they claim that these metrics measure "our understanding" of the mid-depth ocean circulation, in my opinion, they are more consistent with some 10-days prediction skills of the models. It should be pointed out that prediction skills and the ability of a model to resolve some particular physical processes (which would really be a measure of our understanding) are two quite different things. An excellent submesoscale-resolving regional model with no data assimilation is able to reproduce realistic circulation statistics (say EKE, mean flow, dispersion properties etc.), although it will be unable to predict the local current and hydrography on a particular date and time. On the other hand, a coarse resolution data-assimilating system, will be able to decently forecast currents and hydrography (where enough data is available to constrain the model, i.e. near the surface). In the latter case, the model "poorly understands" the ocean, since it does not resolve well the physics, while in the former, the model "understands well" the ocean, since its characteristics are statistically well described. Yet, the poor-resolution data-assimilating model would show better skills at the Lagrangian test the authors propose here. The methods used in this manuscript are only able to measure prediction skill, not to assess the model's ability to reproduce realistic phenomena (including particle dispersion).

Beyond this semantic issue, I see other limitations to this methods: The average time between two profiles is of 10 days and the average velocity at 1000 dbar is of 3.6 cm/s (Ollitruault and Rannou, 2013). The mean displacement is of about 30 km. It seems to me that in any case, a particle advected in a 10 to 20 km resolution velocity field can not successfully reproduce the real-ocean observed displacement. Even if the velocity field was the true velocity interpolated on a coarse grid. During a whole parking phase, the virtual particle will only be visiting 1 to 3 grid points of the model.

Moreover, the duration of the parking phase of an ARGO float (overwhich comparison is made) is of the same order of magnitude as the Lagrangian time scale (Ollitruault and Colin de Verdiere, 2014), which is "the time scale over which the Lagrangian velocities are correlated and is a basic indicator of Lagrangian predictability" (LaCasce 2018). This roughly means that during the time integration of the particle in the model, the velocity at the ARGO float location in the ocean is already completely decorrelated from its original value.

It seems to me that, with the methodology used here, the failure of models to correctly predict particle trajectories is inevitable and can be deduced from simple scaling

arguments.

In my opinion, comparing statistical properties of ARGO floats absolute dispersion with large clusters of virtual particles would be a more straightforward and reliable approach (assessing the difference between statistics, rather than computing statistics of the differences). So would be directly comparing mean Eulerian velocity.

Minor comments :

Throughout the text: I would not use the term "understood", or "understanding", but rather "well forecast" or "accurately modeled". Processes occurring in some region of the ocean can be well understood (through observational or process studies), yet poorly modeled for a variety of reason (poor resolution, in particular below the thermocline; lack of assimilated observations ; inadequate diffusivity coefficients or advection schemes etc.). On the other hands, some regional models may reproduce with impressive skills some observed ocean processes whose physics may remain poorly understood.

Lines (5-6) : The sentence sounds somewhat strange. The ocean circulation is indifferent to models. Could you rephrase ?

Lines 26 : Isn't heat redistributed by the circulation precisely because water masses are? You may want to rephrase this sentence.

Line 27 : "vertical mixing" is explicit enough and "exchange" seems vague and redondent.

Lines 28-29: I don't understand what you mean here. Could you reword this sentence ?

Lines 29-31 : What do you mean by "Depth-related" processes of interaction? Could you reword this sentence ?

Line 32 : What do you mean by "altering the flow field direction, velocity ..."? Isn't the ocean circulation precisely the flow field?

Lines 47-49 : I believe this claim is not accurate. Ollitrault and Colin de Verdiere (2014), and Ollitrault and Rannou (2013)'s work (which you cite) certainly represents the first detailed assessment of the ocean circulation near 1000 m depth. Your work would rather be the first assessment of global reanalysis and models skills.

Line 70 : The choice of 30 degrees for DOD and 50 % for PDOV to discriminate between "well-understood" (which, in my opinion should be changed to "well-modelled") seems arbitrary. 50% sounds very large to me and some explanation about this choice is needed.

Lines 197-199: I don't think your study really proves this. You might want to cite some previous work to make your point here.

Lines 200-202 : references 39-41 discuss diapycnal mixing and small-scale turbulence. They are not related with the isopycnal turbulence (O(1-50) km that is unresolved by the models you tested and might explain the discrepancies with observations.)

Line 204 : What is an Ocean with weak amplitude?

Line 206-207: This statement makes sense, and somehow highlights the limits of the methods you use. In a turbulent ocean with considerable variance in the submesoscale to mesoscale bands, poor skills of coarse-resolution global models is expected. It seems expected and inevitable that the results of a comparison between modelled and observed trajectories on a 10 days time scale (hence a 50-100 km length scale) would conclude to poor model skills at these scales.

Lines 216-219 : I definitely agree with this statement. I am afraid that the implications is that this study is showing a result that is somewhat expected from simple scaling arguments.

Line 387-388 : I am not sure of the current standard, and that depends a lot of the local Rossby radius, but it seems to me that 18 km is "eddy-permitting" rather than "eddy-resolving".

Line 422 : replace layer by depth.

References:

LaCasce, J. (2008). Statistics from Lagrangian observations. Progress in Oceanography, 77(1), 1-29.

Ollitrault, M., & De Verdière, A. C. (2014). The ocean general circulation near 1000-m depth. Journal of Physical Oceanography, 44(1), 384-409

Ollitrault, M., & Rannou, J. P. (2013). ANDRO: An Argo-based deep displacement dataset. Journal of Atmospheric and Oceanic Technology, 30(4), 759-788.

Reviewer #2 (Remarks to the Author):

I think this is a timely study of ocean flows (mostly related to the equatorial Pacific and the Antarctic Circumpolar Current) that highlights substantial disagreements between what is measured and what is predicted by ocean circulation models. More precisely, the indicators of differences are DOD (difference of direction), DOV (difference of velocity), PDOV (percentage of velocity difference), SD (separation distance), SS (Skill Score based on the cumulative simulated separation distance normalized by the cumulative observed displacement).

Before deciding on the publication, I would like to have the following issues addressed:

1) Line 57: concerning the wording "how much is known about spatial variations" it is known that there are flows (like the Equatorial Undercurrent (EUC) and ACC that exhibit a preferred West-East azimuthal direction. More precisely, detailed mathematical analyses of the EUC and ACC have been performed in which solutions flows were derived:

A. Constantin and R. S. Johnson. An exact, steady, purely azimuthal equatorial flow with a free surface. J. Phys. Oceanography. 46 (2016), no.6, 1935-1945.

A. Constantin and R. S. Johnson. An exact, steady, purely azimuthal flow as a model for the Antarctic Circumpolar Current. J. Phys. Oceanography. 46 (2016), no. 12, 3585-3594.

C. I. Martin. Azimuthal equatorial flows in spherical coordinates with discontinuous stratification. *Physics of Fluids*, 33 (2021), no. 2, 026602.

These previously mentioned studies derive exact solutions (to the nonlinear governing equations) written (in spherical coordinates) that describe azimuthal flows (thus suitable for the description of EUC and ACC) whose velocity varies arbitrarily with depth. The solutions present, at times discontinuous stratification, (that leads to the emergence of internal waves) and provide relations between the pressure at the surface and the resulting distortion.

I would be curious to know whether the measured velocities in this study confirm or disprove the statement that EUC and ACC display a (nearly) azimuthal propagation direction. A statement regarding this aspect would be very much appreciated.

2) About the comment "improved velocity field magnitude and direction data are needed for more than 96% of the global ocean" (at line 76)

it is not clear to me (and perhaps to a potential reader as well) what is the source of error in the models that are mentioned in the paper. What stands at the basis of these models? Are these models data-driven or are they accompanied by an explanatory theory based on differential equations?

3) Apart from signaling disparities between observed velocities and velocities predicted by various existing models, what can be done to ensure that future ocean circulation models would more faithfully represent observed quantities of ocean flows: velocity, pressure distribution? O a general note: could the authors comment on what is the role of a systematic theoretical approach in the understanding of ocean dynamics?

Responses to Reviewer

We are grateful to the Reviewers for your helpful and constructive feedback. We have addressed every comment carefully and revised the manuscript accordingly. Below please find our point-by-point responses to each review comment. *The review comments are highlighted in blue*, while our response is in black. Track-changes completed in the revised manuscript are shown in purple.

Responses to Reviewer #1:

The manuscript “Widespread global disparities between modelled and observed mid-depth ocean currents” aims at assessing the skills of global ocean currents reanalysis near 1000 dbar, by comparing the observed displacements of ARGO floats with trajectories of virtual particles integrated in the modelled velocity fields.

Response [1]:

Thank you so much for your constructive and encouraging comments.

Although the topic tackled here (the poor skills of coarse-resolution models at depth) is of importance, in my opinion, the methods used here are not appropriate and I can not recommend publication. I would however like to insist on the importance of assessing global circulation model skills, particularly in or below the thermocline, and to encourage the authors to pursue their efforts in that direction.

Response [2]:

We greatly appreciate your feedback for capturing the ambition and importance of our study. In this revision, we have made a series of adjustments to strengthen the robustness of the results as requested. We have also provided additional supplementary analyses to increase the transparency of improvements for the method. Please kindly refer to the point-by-point responses below for the specific revisions we have made to the original manuscript.

My major objections about the methods, as well as a list of minor comments are provided below.

Major comment:

The authors compare ARGO float displacements with virtual particles integrated in modelled velocity fields for each single available float displacements. They then compute statistical properties of the angle and velocity magnitude differences.

Although they claim that these metrics measure “our understanding” of the mid-depth ocean circulation, in my opinion, they are more consistent with some 10-days prediction skills of the models. It should be pointed out that prediction skills and the ability of a model to resolve some particular physical processes (which would really be a measure of our understanding) are two quite different things. An excellent submesoscale-resolving regional model with no data assimilation is able to reproduce realistic circulation statistics (say EKE, mean flow, dispersion properties etc.), although it will be unable to predict the local current and hydrography on a particular date and time. On the other hand, a coarse resolution data-assimilating system, will be able to decently forecast currents and hydrography (where enough data is available to constrain the model, i.e. near the surface). In the latter case, the model “poorly understands” the ocean, since it does not resolve well the physics, while in the former, the model “understands well” the ocean, since its characteristics are statistically well described. Yet, the poor-resolution data-assimilating model would show better skills at the Lagrangian test the authors propose here. The methods used in this manuscript are only able to measure prediction skill, not to assess the model’s ability to reproduce realistic phenomena (including particle dispersion).

Response [3]:

Thank you so much for pinpointing the scope of the ability of ocean circulation models to reproduce oceanic processes in detail. The insightful concern led us to think more deeply. We would like to make it clear that, although utilizing data-assimilation optimization exactly as you mentioned, the ocean circulation reanalysis systems selected here all adopt classic and well-recognized OGCMs (Oceanic General Circulation Models), such as the MITgcm (Dimitris, Jean-Michel et al. 2008), the MOM3 (Masumoto, Sasaki et al. 2004), and the NEMO3.1 (Lellouche, Greiner et al. 2018). And the computational framework and parameterization schemes of these models are the results of long-term optimization and step-by-step status stabilization. And in fact, the data-assimilation could correct the models’ initial parameterized configuration (also the initial guess to resolve the physics) with all the available in situ and satellite observations producing analyses.

It is generally believed that these models could reproduce the global circulation status as a whole, especially for dynamic processes above the mesoscale (Fox-Kemper, Adcroft et al. 2019). And for the ability to reproduce the sub-mesoscale processes, they still cannot be compared to the regional model due to the high variability. Additional comparison experiments in the Gulf Stream could confirm this. We found evident differences in kinetics characteristics between simulated ROMS floats and simulated OGCM floats, and an evident lack of EKE. More details are shown in the **Response [4]** and **Response [7]**. The significant lack of energy is what we are attempting to point out. In addition, following your suggestion we have added more statistical analysis for circulation characteristics and further descriptions are shown in the **Response [7]**.

Beyond this semantic issue, I see other limitations to this methods: The average time between two profiles is of 10 days and the average velocity at 1000 dbar is of 3.6 cm/s (Ollitrault and Rannou, 2013). The mean displacement is of about 30 km. It seems to me that in any case, a particle advected in a 10 to 20 km resolution velocity field can not successfully reproduce the real-ocean observed displacement. Even if the velocity field was the true velocity interpolated on a coarse grid. During a whole parking phase, the virtual particle will only be visiting 1 to 3 grid points of the model.

Response [4]:

Thank you so much for the under-powered limitations you raised, about whether coarse velocity fields powered by ocean models could be used to reproduce oceanic processes, is of importance. It is a key point that we neglected before and we appreciate your comments. As this concern is critical, we make three measures to address this concern essentially (details are attached below this summary). We have improved the Methods section to make it more clear how we design the methods and what issues they can help address.

- 1) We made a kinetics comparison between the simulated floats driven by ROMS and OGCMs in the Gulf Stream.
- 2) We add stochastic processes in Lagrangian simulation to improve the ability of models to reproduce high-frequency realistic phenomena.
- 3) We employed particle clustering schemes with more robustness, and additional tests ensure the transparency of the simulation experiment.

The detailed revisions made for each point are as follows:

- 1) Kinetics comparison test (These contents are included in the Methods of the revised manuscript). Details of the test refer to Koszalka (2013) and van Sebille (2018).

“To address potential underpowering below the mesoscale of global ocean circulation models, velocity fields output from the DopAnV3R3-ini2007 (a ROMS-based reanalysis), is used to compare with the velocity fields output from global ocean circulation models. Lagrangian simulation analysis mentioned in the next section is used here. The simulation traverses the Gulf Stream zone and covers 2014-2018, viz. the five years that exhibit the largest Argo displacements, while no diffusivity is set here. For each model, 1,102 simulated floats (equal to displacements number of displacements available to Argo) are released. The velocity comparison result between simulated floats advected by the ROMS model and global models is shown in Supplementary Fig. 6.”

Which ROMS velocity field we used for the Kinetics comparison test is described in the Methods section, as following.

“Regional ocean circulation model velocity dataset

Regional velocity fields used for the kinetics comparison test are provided by the DopAnV3R3-ini2007⁵⁷ (Doppio Analysis Version 3 Release 3). The DopAnV3R3-ini2007 is a ROMS-based (Regional Ocean Modeling System) data assimilative reanalysis focusing on the Mid-Atlantic Bight and Gulf of Maine regions of the northwestern North Atlantic. The reanalysis uses four-dimensional variational (4D-Var) data assimilation (DA) derived from comprehensive observational data from in situ platforms, coastal radars, and satellites⁵⁸. The velocity fields output from the DopAnV3R3-ini2007 (available online <https://www.seanoe.org/data/00785/89673/>) is provided on a 7-km horizontal grid within 40 vertical levels from the coastal ocean to deep sea and covers the period 2007-2021. ”

The Gulf Stream is one of the intense western boundary currents which are of the most striking feature of the ocean circulation, serving to transport heat poleward as they carry warm tropical and subtropical waters to higher latitudes. In the region of the Gulf Stream, intensive oceanographic surveys and observation projects are carried out here. These are the reason we chose the region for the test.

Supplementary Fig. 6 | The probability density function (PDF) of absolute velocity from simulated floats without diffusivity. All floats are deployed only in the Gulf Stream. The grey line represents the simulated floats advected by ROMS velocity fields. Other colored lines represent the simulated floats advected by OFES, ECCO2, CMEMS GLORYS12 velocity fields, respectively.

- 2) Stochastic diffusivity test (These contents are also included in the Methods of the revised manuscript)”

“As emphasized in a series of other studies, the diffusivity of Lagrangian analysis is a significant issue⁵⁹⁻⁶¹. In general, stochastic term is used in Lagrangian analysis to model stochastic fluctuations including sub-grid scale diffusion and unresolved physics such as eddies, waves, turbulence or mixing processes. A map of diffusivity may be estimated by floats observations^{62,63}. However, different diffusivities have been set up for ocean general circulation models^{48,49}. So here horizontal isotropic diffusivities account for the dynamic

processes that are not fully resolved by the global ocean circulation models³⁷. The stochastic walk of the particle tracking is implemented by the Wiener noise process.

To quantify the suitable diffusivities, a stochastic diffusivity test was conducted by Lagrangian simulation only in the Gulf Stream zone. As previous studies noted, validations of high-resolution model simulations are essential to ensure applicability of the stochastic term^{60,64}. We conduct the test through comparing the Lagrangian simulation results between global models and ROMS. No diffusivity was set for ROMS simulation.

Supplementary Fig. 7 visualizes the changes of kinetics indicating by absolute velocity for each model when the diffusivity varies. For simulations of ECCO2 and OFES, their kinetic distributions gradually approach that of ROMS as the diffusivity increases, with the frequency tending to decrease for small velocities ($<10 \text{ cm s}^{-1}$) and increase for large velocities ($>10 \text{ cm s}^{-1}$). It seems that the applicable value is around 1000 to $1500 \text{ m}^2 \text{ s}^{-1}$. Further tests, detailed below, may provide accurate values. Instead, the kinetic distributions of CMEMS GLORYS12 simulations moves away from that of ROMS as the diffusivity leaves zero. Accordingly, the stochastic diffusivity is set to $10^{-5} \text{ m}^2 \text{ s}^{-1}$ which is indeed close to zero⁶⁶. It makes sense for a non-zero diffusivity which allows floats to deviate from deterministic trajectories driven by sub-grid scale diffusion.

To further test the diffusivity value for ECCO2 and OFES, correlations of probability density functions of velocities are calculated as shown in Supplementary Fig. 8. Linear correlations of each simulation of different diffusivity with the ROMS simulation are calculated. According to the maximum correlation, the diffusivity for ECCO2 is $1100 \text{ m}^2 \text{ s}^{-1}$ and the value for OFES is $1400 \text{ m}^2 \text{ s}^{-1}$. The diffusivities here demonstrated reasonable agreement with the global average value estimated by Cole et al.⁶⁵. Moreover, the Kolmogorov-Smirnov test results show that the simulated kinetic distribution is consistent with that of the observed kinetics.

References:

- Cole, S. T., Wortham, C., Kunze, E. and Owens, W. B. (2015). Eddy stirring and horizontal diffusivity from Argo float observations: Geographic and depth variability. *Geophysical Research Letters*, 42(10): 3989-3997.
- De Dominicis, M., Leuzzi, G., Monti, P., Pinardi, N. and Poulain, P.-M. (2012). Eddy diffusivity derived from drifter data for dispersion model applications. *Ocean Dynamics*, 62(9): 1381-1398.
- Roach, C. J., Balwada, D. and Speer, K. (2016). Horizontal mixing in the Southern Ocean from Argo float trajectories. *Journal of Geophysical Research: Oceans*, 121(8): 5570-5586.
- Hunter, J., Craig, P. and Phillips, H. (1993). On the use of random walk models with spatially variable diffusivity. *Journal of Computational Physics*, 106(2): 366-376.

Koszalka, I., LaCasce, J. H. and Mauritzen, C. (2013). In pursuit of anomalies—analyzing the poleward transport of Atlantic Water with surface drifters. *Deep Sea Research Part II: Topical Studies in Oceanography*, 85: 96-108.

LaCasce, J. H. (2008). Statistics from Lagrangian observations. *Progress in Oceanography*, 77(1): 1-29.

Lellouche, J. M., Greiner, E., Le Galloudec, O., Garric, G., Regnier, C., Drevillon, M., Benkiran, M., Testut, C. E., Bourdalle-Badie, R., Gasparin, F., Hernandez, O., Levier, B., Drillet, Y., Remy, E. and Le Traon, P. Y. (2018). Recent updates to the Copernicus Marine Service global ocean monitoring and forecasting real-time 1/12° high-resolution system. *Ocean Sci.*, 14(5): 1093-1126.

Masumoto, Y., Sasaki, H., Kagimoto, T., Komori, N., Ishida, A., Sasai, Y., Miyama, T., Motoi, T., Mitsudera, H., Takahashi, K., Sakuma, H. and Yamagata, T. (2004). A fifty-year eddy-resolving simulation of the World Ocean - preliminary outcomes of OFES (OGCM for the Earth Simulator). *Journal of the Earth Simulator*, 1: 35-56.

van Sebille, E., Griffies, S. M., Abernathey, R., Adams, T. P., Berloff, P., Biastoch, A., Blanke, B., Chassignet, E. P., Cheng, Y., Cotter, C. J., Deleersnijder, E., Döös, K., Drake, H. F., Drijfhout, S., Gary, S. F., Heemink, A. W., Kjellsson, J., Koszalka, I. M., Lange, M., Lique, C., MacGilchrist, G. A., Marsh, R., Mayorga Adame, C. G., McAdam, R., Nencioli, F., Paris, C. B., Piggott, M. D., Polton, J. A., Rühs, S., Shah, S. H. A. M., Thomas, M. D., Wang, J., Wolfram, P. J., Zanna, L. and Zika, J. D. (2018). Lagrangian ocean analysis: Fundamentals and practices. *Ocean Modelling*, 121: 49-75.

Wang, T., Du, Y. and Wang, M. (2022). Overlooked current estimation biases arising from the Lagrangian Argo trajectory derivation method. *Journal of Physical Oceanography*, 52(1): 3-19.

Wang, T., Gille, S. T., Mazloff, M. R., Zilberman, N. V. and Du, Y. (2020). Eddy induced acceleration of Argo floats. *Journal of Geophysical Research: Oceans*, 125(10).

Supplementary Fig. 7 | The probability density function (PDF) of absolute velocity from stochastic float clusters with different diffusivity configurations. (a) for ECCO2 simulation, (b) for OFES simulation, (c) for CMEMS GLORYS12 simulation. All floats are deployed only in the Gulf Stream. The red line represents the simulated floats advected by ROMS velocity fields with no diffusivity. Other colored dashed lines represent the simulated floats with diffusivity at 0-3000 m^2s^{-1} , respectively.

Supplementary Fig. 8 | The Correlation of the displacements.

3) Spatial coverage test for the size of cluster.

“This Lagrangian simulation allows the generation of float clusters, improving the statistical significance of the outputs. A number of virtual floats is released at the starting position of each Argo cycle. The centroid of the displacement probability density cloud is considered to be representative of the float cluster displacement. The size of cluster (that is, the number of virtual floats for each release) is determined by a spatial coverage test as shown in Supplementary Fig. 9. We selected the velocity field of OFES as it has the largest diffusivity among the models, thereby ensuring the robustness of the value. Given the large number of simulations performed, we selected 100 as the size of cluster to balance statistical accuracy and computational intensity.”

Supplementary Fig. 9 | The convergence test of spatial coverage after simulations completed. The spatial coverage varies with the size of each cluster, indicated by the number of grids on which the simulated floats are located. The simulations of the OFES model in the Gulf Stream are taken as the standard. The size of the grid unit ($1/100^\circ \times 1/100^\circ$) is one-tenth of the horizontal resolution of the OFES velocity field. The blue stars represent results from 10 simulation cases, and the red line indicates polynomial fit.

Moreover, the duration of the parking phase of an ARGO float (overwhich comparison is made) is of the same order of magnitude as the Lagrangian time scale (Ollitrault and Colin de Verdiere, 2014), which is “the time scale over which the Lagrangian velocities are correlated and is a basic indicator of Lagrangian predictability” (LaCasce 2018). This roughly means that during the time integration of the particle in the model, the velocity at the ARGO float location in the ocean is already completely decorrelated from its original value.

Response [5]:

Thank you so much for your comments. Indeed, the Lagrangian trajectory simulation of Argo floats have bias that grows with time, as you stated. As you have concerned, previous studies show that the simulated floats with long simulation time tended to perform larger biases. Yet, the numerical experiments conducted by Wang, Du et al. (2022) illustrated that the effective time scale for Argo temporal sampling is less than 20 days (longer than the 10-day set in our study). Please see [FIG.11] and [FIG. C1] in Wang, Du et al. (2022). Therefore, we believe that the Lagrangian trajectory simulation of Argo float for around 10 days in this study is reasonable and the related biases are acceptable.

[redacted]

[FIG.11] in Wang, Du et al. (2022). Mean velocities derived from the trajectories of simulated floats (black) that are configured in various parking stages (i.e., 1, 2, 5, 10, 20, and 50 days). As counterparts, mean theoretical velocities of simulated floats in the ECCO2 are overlapped (red).

[redacted]

[FIG. C1] in Wang, Du et al. (2022). Spatial distribution of velocities in the ACC derived from the trajectories of simulated floats that are configured in various parking stages (i.e., 1, 2, 5, 10, 20, and 50 days). Black dots in (e) and (f) denote regions where the derived velocities are distorted, where the mean speed is less than 60% of the ECCO2 climatology.

References:

Wang, T., Du, Y. and Wang, M. (2022). Overlooked current estimation biases arising from the Lagrangian Argo trajectory derivation method. *Journal of Physical Oceanography*, 52(1): 3-19. DOI: <https://doi.org/10.1175/JPO-D-20-0287.1>.

It seems to me that, with the methodology used here, the failure of models to correctly predict particle trajectories is inevitable and can be deduced from simple scaling arguments.

Line 206-207: This statement makes sense, and somehow highlights the limits of the methods you use. In a turbulent ocean with considerable variance in the submesoscale to mesoscale bands, poor skills of coarse-resolution global models is expected. It seems expected and inevitable that the results of a comparison between modelled and observed trajectories on a 10 days time scale (hence a 50-100 km length scale) would conclude to poor model skills at these scales.

Lines 216-219 : I definitely agree with this statement. I am afraid that the implications is that this study is showing a result that is somewhat expected from simple scaling arguments.

Response [6]:

Thank you so much for your comments. We agree with you that the resolution of ocean models does lead to poor prediction skills. The simulation of CMEMS has better performance than that of ECCO2 and OFES (see Fig. 2). We make suggestions in the Discussion section for appealing to improve the resolution of ocean global circulation models.

According to your guide, we have done a longer timescale (100-day) Lagrangian experiments to confirm the impact of timescales on poor predictability, as the Supplementary Fig. 10 shown. Considering that the error of the random diffusivity term is time-dependent, no stochastic diffusivity is set in this simulation (Roach, Balwada et al. 2016, Wang, Gille et al. 2020). As we can see, the prediction skill of the seasonal timescale has improved somewhat and is still widely inadequate.

Fig. 2. Global spatial distribution of DOD and DOV for ECCO2 (a, b), OFES (c, d), CMEMS (e, f). Left panels: mean DOD during 2001–2020 (shaded color). Warm (Cool) color indicates that observed velocity of float displacements exceeds (lags) simulated velocity. Right panels: averaged DOV during 2001–2020 (shaded color).

Supplementary Fig. 10 | Global spatial distribution of (a) DOD and (b) DOV for OFES in 100-day simulations. The red and blue colors in upper panel represent the highest positive and negative DOV, respectively. The red and blue colors in under panel represent the highest and lowest DOD (only positive).

References:

Roach, C. J., Balwada, D. and Speer, K. (2016). Horizontal mixing in the Southern Ocean from Argo float trajectories. *Journal of Geophysical Research: Oceans*, 121(8): 5570-5586.

Wang, T., Gille, S. T., Mazloff, M. R., Zilberman, N. V. and Du, Y. (2020). Eddy induced acceleration of Argo floats. *Journal of Geophysical Research: Oceans*, 125(10).

In my opinion, comparing statistical properties of ARGO floats absolute dispersion with large clusters of virtual particles would be a more straightforward and reliable approach (assessing the difference between statistics, rather than computing statistics of the differences). So would be directly comparing mean Eulerian velocity.

Response [7]:

We agree with you that comparing statistical properties are of importance to refine our study. Following your kind advice, we have added more substantial analysis on statistical circulation characteristics in the Results section. It is worth mentioning that statistical Eulerian fields derived from Lagrangian float trajectories could be biased due to the strong eddy-jet interactions acceleration (Wang, Du et al. 2022). The revised analyses are as following:

“Significant kinetic deviations exist in zonal and meridional velocity (Supplementary Fig. 11) and in the mean velocity fields (Supplementary Fig. 12). These systematic biases may be caused by lack of eddy kinetic energy (Supplementary Fig. 13). Significant underestimation of eddy kinetic energy is observed in almost all the western boundary current regions.”

Supplementary Fig. 11 | The probability density function (PDF) of the zonal velocity and meridional velocity from Argo floats and simulated float clusters. (a) Zonal velocity (b) Meridional velocity. The red line represents Argo floats, and the magenta, green, blue line represent simulated floats of ECCO2, OFES, CMEMS GLORYS12, respectively.

Supplementary Fig. 12 | Global spatial distribution of mean velocity calculated from (a) Argo floats (b) OFES velocity field. The red (blue) color indicates high (low) velocities.

Supplementary Fig. 13 | Global spatial distribution of eddy kinetic energy calculated from (a) Argo floats (b) Simulated OFES floats. The red and blue color indicate high and low eddy kinetic energy distributions, respectively.

Minor comments :

Response [8]:

Thank you so much for your careful reading. All the points mentioned are either corrected or properly explained in the revised manuscript. See below for the corresponding edits.

Throughout the text: I would not use the term “understood”, or “understanding”, but rather “well forecast” or “accurately modeled”. Processes occurring in some region of the ocean can be well understood (through observational or process studies), yet poorly modeled for a variety of reason (poor resolution, in particular below the thermocline; lack of assimilated observations ; inadequate diffusivity coefficients or advection schemes etc.). On the other hands, some regional models may reproduce with impressive skills some observed ocean processes whose physics may remain poorly understood.

Response [9]:

We appreciate you for the valuable comments. To be more rigorous, in this revised manuscript, we use the term “*accurately modelled*” or “*improved prediction skills*” on assessment of prediction skills for the models. We revise the entire manuscript, particularly all the related terminologies, to accurately define what we are confident about measuring.

Lines (5-6) : The sentence sounds somewhat strange. The ocean circulation is indifferent to models. Could you rephrase ?

Lines 26 : Isn't heat redistributed by the circulation precisely because water masses are? You may want to rephrase this sentence.

Line 27 : “vertical mixing” is explicit enough and “exchange” seems vague and redundant.

Lines 28-29: I don't understand what you mean here. Could you reword this sentence?

Lines 29-31: What do you mean by “Depth-related” processes of interaction? Could you reword this sentence?

Response [10]:

Above points are all revised accordingly.

The first paragraph of the revised Introduction section is:

“The oceans, covering more than 70% of the earth's surface and containing 95% of all water, critically modulate climate change through carbon dioxide sequestration and by absorbing excess heat¹⁻¹¹. Dynamically, the large-scale ocean circulation plays a key role in the Earth's climate system by redistributing heat of the ocean^{12,13}. Stratification arises through the processes of vertical mixing, one of the products of which is the mid-depth

ocean near 1,000 m between variable waters near the surface and stable waters at greater depth¹⁴. Interactions between surface currents and thermohaline currents inject changes into the mid-depth ocean near 1,000 m. In turn, mid-depth ocean circulation has a significant impact on the global ocean as a whole by altering the physical characteristics of ocean waters¹⁵⁻¹⁸ that play a fundamental role in the longer-term evolution of Earth's climate system. However, knowledge of mid-depth ocean circulation is hampered by the lack of direct observations and novel methods are therefore needed to improve data, modeling accuracy, and mechanism of mid-depth ocean circulation and related processes.”

Line 32: What do you mean by “altering the flow field direction, velocity ...”? Isn't the ocean circulation precisely the flow field?

Response [11]:

Have been rephrased. The previous semantic confusion has been corrected, like:

“In turn, mid-depth ocean circulation has a significant impact on the global ocean as a whole by altering the physical characteristics of ocean waters¹⁵⁻¹⁸ that play a fundamental role in the longer-term evolution of Earth's climate system.”

Lines 47-49 : I believe this claim is not accurate. Ollitrault and Colin de Verdiere (2014), and Ollitrault and Rannou (2013)'s work (which you cite) certainly represents the first detailed assessment of the ocean circulation near 1000 m depth. Your work would rather be the first assessment of global reanalysis and models skills.

Response [12]:

We have revised this point and we are cautious not to overclaim that we have assessed the ocean circulation. The revised statement is:

“Using comprehensive and quantitative methods, we present here the first detailed assessment and validation of global ocean circulation model skills near 1,000 m depth.”

Line 70 : The choice of 30 degrees for DOD and 50 % for PDOV to discriminate between “well-understood” (which, in my opinion should be changed to “well-modelled”) seems arbitrary. 50% sounds very large to me and some explanation about this choice is needed.

Response [13]:

Thank you so much for your thoughtful comments.

The metrics we designed like DOD and PDOV, are some of the most basic ones to depict floats discrepancy in order to ensure the transparency of the analysis. As the

analysis demonstrates, the discrepancies between observation and modelling are indeed higher than would normally be expected. The reasonableness and robustness of the threshold choice is also verified in the Methods section of the "statistics of several composite parameters" part. The classified area with well-modelled ocean does not change significantly as the threshold choice varies.

Lines 197-199: I don't think your study really proves this. You might want to cite some previous work to make your point here.

Lines 200-202 : references 39-41 discuss diapycnal mixing and small-scale turbulence. They are not related with the isopycnal turbulence (O(1-50) km that is unresolved by the models you tested and might explain the discrepancies with observations.)

Response [14]:

We have rephased the potential factors of the modelled discrepancies as following:

“Of particular importance is the finding that circulation energy is underestimated across almost all of the global oceans. This arises in part because high-frequency dynamics are poorly resolved in ocean circulation models and also due to the inadequate solutions of sub-grid processes, such as the lack of eddy kinetic energy illustrated in Supplementary Fig. 13. Moreover, parameterizations of sub-grid processes result in a series of errors, a situation which remains still the most advanced challenge in ocean modelling^{43,60}. Taking tidal processes as an example, previous studies have confirmed that tidal currents traversing rough bathymetry could induce turbulence and mixing to accelerate mid-depth ocean circulation³⁸⁻⁴⁰. ”

The small-scale turbulence motions we discuss here are intended to explore possible negligence in models' parameterizations.

Line 204 : What is an Ocean with weak amplitude?

Response [15]:

Have been rephrased. What we would like to express is that oceans with slower mean velocity.

Line 206-207: This statement makes sense, and somehow highlights the limits of the methods you use. In a turbulent ocean with considerable variance in the submesoscale to mesoscale bands, poor skills of coarse-resolution global models is expected. It seems expected and inevitable that the results of a comparison between modelled and observed trajectories on a 10 days time scale (hence a 50-100 km length scale) would conclude to poor model skills at

these scales.

Response [16]:

We hope this concern could be answered by the improved methods proposed for the spon [4].

Lines 216-219 : I definitely agree with this statement. I am afraid that the implications is that this study is showing a result that is somewhat expected from simple scaling arguments.

Response [17]:

Thanks for the careful attention. In this study, the prediction skills of CMEMS GLORYS12 are better than the other 2 models. But actually, the models have similar temporal and spatial resolutions for velocity field outputs, as shown in the following:

- a) ECCO2: -day average and $1/4^\circ$ horizontal resolution;
- b) OFES: 1-day average and $1/1^\circ$ horizontal resolution;
- c) CMEMS GLORYS12: 1-day average and $1/12^\circ$ horizontal resolution.

So we hope the scaling problem could be addressed largely by the improved methods.

Line 387-388 : I am not sure of the current standard, and that depends a lot of the local Rossby radius, but it seems to me that 18 km is “eddy-permitting” rather than “eddy-resolving”.

Line 422 : replace layer by depth.

Response [18]:

Above points are all revised accordingly.

References:

LaCasce, J. (2008). Statistics from Lagrangian observations. Progress in Oceanography, 77(1), 1-29.

Ollitrault, M., & De Verdière, A. C. (2014). The ocean general circulation near 1000-m depth. Journal of Physical Oceanography, 44(1), 384-409

Ollitrault, M., & Rannou, J. P. (2013). ANDRO: An Argo-based deep displacement dataset. Journal of Atmospheric and Oceanic Technology, 30(4), 759-788.

Response [19]:

All of the studies listed here are very instructive and have been referred.

Hope the point-by-point responses above have addressed all concerns. We appreciate for all your positive comments and suggestions!

Responses to Reviewer #2:

I think this is a timely study of ocean flows (mostly related to the equatorial Pacific and the Antarctic Circumpolar Current) that highlights substantial disagreements between what is measured and what is predicted by ocean circulation models. More precisely, the indicators of differences are DOD (difference of direction), DOV (difference of velocity), PDOV (percentage of velocity difference), SD (separation distance), SS (Skill Score based on the cumulative simulated separation distance normalized by the cumulative observed displacement).

Response [1]:

Thank you so much for the constructive and encouraging comments.

Before deciding on the publication, I would like to have the following issues addressed:

1) Line 57: concerning the wording "how much is known about spatial variations" it is known that there are flows (like the Equatorial Undercurrent (EUC) and ACC that exhibit a preferred West-East azimuthal direction. More precisely, detailed mathematical analyses of the EUC and ACC have been performed in which solutions flows were derived:

A. Constantin and R. S. Johnson. An exact, steady, purely azimuthal equatorial flow with a free surface. J. Phys. Oceanography. 46 (2016), no.6, 1935-1945.

A. Constantin and R. S. Johnson. An exact, steady, purely azimuthal flow as a model for the Antarctic Circumpolar Current. J. Phys. Oceanography. 46 (2016), no. 12, 3585-3594.

C. I. Martin. Azimuthal equatorial flows in spherical coordinates with discontinuous stratification. Physics of Fluids, 33 (2021), no. 2, 026602.

These previously mentioned studies derive exact solutions (to the nonlinear governing equations) written (in spherical coordinates) that describe azimuthal flows (thus suitable for the description of EUC and ACC) whose velocity varies arbitrarily with depth. The solutions present, at times discontinuous stratification, (that leads to the emergence of internal waves) and provide relations between the pressure at the surface and the resulting distortion.

I would be curious to know whether the measured velocities in this study confirm or disprove the statement that EUC and ACC display a (nearly) azimuthal propagation direction. A statement regarding this aspect would be very much appreciated.

Response [2]:

We appreciate you for the valuable and instructive comments. What is concerned here represents the cutting-edge topic in geophysical fluid dynamics. We are honored to discuss the frontier field.

Argo measurement and previous studies show that velocities of both EUC and ACC vary with depth. Refined vertical structure of EUC were drawn by Busecke et al. (2019) in their [Figure 2] and relatively coarse structure (due to Argo sampling intensity) near

1000-m depth were given by Ollitrault and DeVerdiere (2014) in [FIG.1] which using essentially the same data as our study. Both structures at different depths depict zonal patterns but with huge changes in velocities (more than 30 cm/s above 300 m and less than 5 cm/s near 1,000 m depth).

For ACC, we computed the velocities from Argo floats at 30-90°S, see the following Response Fig.1. Compared to surface velocity, Argo measurement indicate that strong jet streams (more than 15 cm/s on core paths) of ACC weakened but not dramatically from surface to 1,000 m depth. This trend is analogous to the azimuthal flows study and also consistent with previous studies (Tomczak and Godfrey 1994, Firing et al. 2011).

Response Fig. 1 | Spatial distribution of observed velocity from Argo floats at middle to high latitude oceans (30-90°S).

Moreover, the idea that “velocity varies arbitrarily with depth” is comparable to the Beta spirals theory which first proposed by Stommel F. and Schott H. (Stommel and Schott 1977, Schott and Stommel 1978). The theory infers that the velocity tends to rotate with increasing depth in ocean interior. It also diagnoses the rotation with depth is regulated by the vertical velocity and stratification (similar with azimuthal flows study).

All of the studies listed here are pioneering in geophysical fluid dynamics and have been highlighted in the Discussion section of the revised manuscript.

“We would like to point out the recent breakthrough in modelling large-scale ocean currents through exact solutions to the geophysical fluid dynamics governing equations (Constantin and Johnson 2016a, Constantin and Johnson 2016b, Martin and Quirchmayr 2019, Martin 2021, Martin and Quirchmayr 2022). This kind of pioneering mathematical analysis of geophysical currents could lead to series of in-depth theoretical studies (Johnson 2018).”

References:

Busecke, J. J. M., Resplandy, L., & Dunne, J. P. P. (2019). The Equatorial Undercurrent and the oxygen minimum zone in the Pacific. *Geophysical Research Letters*, 46, 6716– 6725. <https://doi.org/10.1029/2019GL082692>.

Ollitrault, Michel and Alain Colin de Verdière. "The Ocean General Circulation near 1000 m Depth." *Journal of Physical Oceanography* 44 (2): 394-409. <https://doi.org/10.1175/JPO-D-13-030.1>

Stommel, H. and Schott, F. (1977). The beta spiral and the determination of the absolute velocity field from hydrographic station data. *Deep Sea Research*, 24(3): 325-329. [https://doi.org/10.1016/0146-6291\(77\)93000-4](https://doi.org/10.1016/0146-6291(77)93000-4).

Schott, F. and Stommel, H. (1978). Beta spirals and absolute velocities in different oceans. *Deep Sea Research*, 25(11): 961-1010. [https://doi.org/10.1016/0146-6291\(78\)90583-0](https://doi.org/10.1016/0146-6291(78)90583-0).

Tomczak, M., and J. S. Godfrey, 1994: *Regional Oceanography: An Introduction*. Pergamon Press, 422 pp.

Firing, Y. L., Chereskin, T. K., and Mazloff, M. R. (2011), Vertical structure and transport of the Antarctic Circumpolar Current in Drake Passage from direct velocity observations, *J. Geophys. Res.*, 116, C08015. <https://doi.org/10.1029/2011JC006999>.

2) About the comment "improved velocity field magnitude and direction data are needed for more than 96% of the global ocean" (at line 76) it is not clear to me (and perhaps to a potential reader as well) what is the source of error in the models that are mentioned in the paper. What stands at the basis of these models? Are these models data-driven or are they accompanied by an explanatory theory based on differential equations?

Response [3]:

We thank a lot for your in-depth concerning. As you mentioned constructively, the errors depend on the model bases which driven by both explanatory theories and data-driven. Actually, ocean models used in this study include both OGCM (Ocean General Circulation Model) and reanalysis. The OGCMs are basically continuum equations of motion for seawater, solved by discretization approximations. The idea of ocean reanalysis is to use direct observations to correct model errors to improve the analysis of ocean state estimates (Carton and Giese 2008).

For details, the equations for oceanic motions have been long known (Navier 1822, Stokes 1845, Onsager 1931). According to the characteristics of large-scale geophysical fluid dynamics, the differential equations are simplified by approximation and discretization to produce finite-differencing solutions. The hydrostatic approximations and the Boussinesq are widely used in approximations. In terms of grid discretization, the setting of resolutions significantly affects the accuracy of vertical discretization and horizontal discretization. Due to computationally intensive acquisition of the large-scale numerical simulations, the sub-grid scale features are described by parameterization, which is the most advanced challenge (Griffies, Adcroft et al. 2009, Fox-Kemper, Adcroft et al. 2019). For the data-assimilation, multi-source observation data used for reanalysis include sea surface temperature, sea surface

salinity, sea ice concentration, sea level anomaly, sea surface height, in-situ temperature and salinity, etc.

Some of the main sources of model errors are discussed here. In summary, errors occurring in ocean circulation models are brought by inadequate numerical algorithms, uncharacteristic sub-grid process parameterizations, and improper representation of initial and boundary conditions. First of all, the physical assumption and approximations can lead to numerical instabilities. Due to the resolution limit and insufficient understanding in certain physics, parameterizations are always used to account for physics and sub-grid processes numerically. But the interaction of various parameterization terms results in complex errors. Furthermore, the initial conditions such as sea temperature, sea salinity, and other features in long-term average (climatology) are adopted to initialize the model, which is always provided by external climatology models. This initial forcing brings errors from external systems.

In the following response, we will discuss the proposals to reduce the errors according to this study.

References:

Carton, J. A. and Giese, B. S. (2008). A reanalysis of ocean climate using Simple Ocean Data Assimilation (SODA). *Monthly Weather Review*, 136(8): 2999-3017.

Navier, C. L. (1822). *Mémoire sur les lois du mouvement des fluides*. *Mem. de l'Acad. des Sci.*,(389): 234–245.

Fox-Kemper, B., Adcroft, A., Böning, C. W., Chassignet, E. P., Curchitser, E., Danabasoglu, G., Eden, C., England, M. H., Gerdes, R., Greatbatch, R. J., Griffies, S. M., Hallberg, R. W., Hanert, E., Heimbach, P., Hewitt, H. T., Hill, C. N., Komuro, Y., Legg, S., Le Sommer, J., Masina, S., Marsland, S. J., Penny, S. G., Qiao, F., Ringler, T. D., Treguier, A. M., Tsujino, H., Uotila, P. and Yeager, S. G. (2019). Challenges and Prospects in Ocean Circulation Models. *Frontiers in Marine Science*, 6.

Griffies, S., Adcroft, A., Banks, H., Böning, C., Chassignet, E., Danabasoglu, G., Danilov, S., Deleersnijder, E., Drange, H. and England, M. (2009). Problems and prospects in large-scale ocean circulation models. *Proceedings of OceanObs*, 9: 410-431.

Onsager, L. (1931). Reciprocal relations in irreversible processes. I. *Physical review*, 37(4): 405.

Riser, S. C., Freeland, H. J., Roemmich, D., Wijffels, S., Troisi, A., Belbéoch, M., Gilbert, D., Xu, J., Pouliquen, S., Thresher, A., Le Traon, P.-Y., Maze, G., Klein, B., Ravichandran, M., Grant, F., Poulain, P.-M., Suga, T., Lim, B., Sterl, A., Sutton, P., Mork, K.-A., Vélez-Belchí, P. J., Ansorge, I., King, B., Turton, J., Baringer, M. and Jayne, S. R. (2016). Fifteen years of ocean observations with the global Argo array. *Nature Climate Change*, 6(2): 145-153.

3) Apart from signaling disparities between observed velocities and velocities predicted by various existing models, what can be done to ensure that future ocean circulation models would more faithfully represent observed quantities of ocean flows: velocity, pressure distribution? O a general note: could the authors comment on what is the role of a systematic theoretical approach in the understanding of ocean dynamics?

Response [4]:

Thank you so much for your suggestion. We have revised the Discussion section to incorporate the outlook:

“As noted by Fox-Kemper, Adcroft et al.⁴³, numerical and parameterization improvements will continue to define the state of the art in ocean circulation modelling. In the future, we expect ocean circulation models to represent observed quantities of ocean flows more faithfully through:

(a) More intensive and qualified observations. There is a pressing need for more observations in the mid-depth ocean. A proposal to improve the sampling density of the Argo program particularly in parts of the world ocean where climate impacts are especially strong, was raised by Riser, Freeland et al.⁴⁴. Here we recommend enhancement of observations in the mid-depth ocean, such as by increasing the float deployment and adding an inertial navigation component in float design which can provide locations at greater frequency during the parking period.

(b) More productive parameterization. The ocean is a non-constant and non-uniform dynamical system with high-frequency variations. Thus, productive parameterization is particularly challenging and much needed. To strengthen the prediction skill of mid-depth ocean currents, the parameterization of global modelling is expected to be improved, such as depth-dependent eddy diffusivity and viscosity^{47,48}, internal mixing with spatio-temporal variability⁴⁹, ice-ocean basal friction^{50,51}.

(c) Finer resolution. With the rapid growth of high-performance computing techniques and cloud-based platforms, higher-resolution global oceanic circulation modelling systems may provide better estimates for certain regions⁵²⁻⁵⁴.

(d) Last but not least, we would like to point out the recent breakthrough in modelling large-scale ocean currents through exact solutions to the geophysical fluid dynamics governing equations⁵⁵⁻⁵⁹. This kind of pioneering mathematical analysis of geophysical currents may prompt further in-depth theoretical studies⁶⁰. ”

As for “the role of a systematic theoretical approach,” the physical-theoretical approaches are of fundamental importance as bases of the dynamic framework and systematic processes of the ocean models.

We appreciate for all your positive comments and suggestions!

References:

Carton, J. A. and Giese, B. S. (2008). A reanalysis of ocean climate using Simple Ocean Data Assimilation (SODA). *Monthly Weather Review*, 136(10): 2999-3017.

Dimitris, M., Jean-Michel, C., Patrick, H., Cornelius, H., T., L., An, S., Michael, S. and H., Z. (2010). ECCO2: high resolution global ocean and sea ice data synthesis. *AGU Fall Meeting Abstracts*, 11.11.11.

Fox-Kemper, B., Adcroft, A., Böning, C. W., Chassignet, E. P., Curchitser, E., Danabasoglu, G., Eden, C., England, M. H., Gerdes, R., Greatbatch, R. J., Griffies, S. M., Hallberg, R. W., Hanert, E., Heimbach, P., Hewitt, H. T., Hill, C., Komuro, Y., Legg, S., Le Sommer, J., Masina, S., Marsland, S. J., Penny, S. G., Qiao, F., Ringler, T. D., Treguier, A. M., Tsujino, H., Uotila, P. and Yeager, S. G. (2019). Challenges and prospects in ocean circulation models. *Frontiers in Marine Science*, 6.11.2019.

Griffies, S., Adcroft, A., Banks, H., Böning, C., Chassignet, E., Danabasoglu, G., Danilov, S., Deleersnijder, E., Drange, H. and England, M. (2009). Problems and prospects in large-scale ocean circulation models. *Proceedings of OceanObs*, 9: 41-44.

Lellouche, J. M., Greiner, E., Le Galloudec, O., Garric, G., Regnier, C., Drevillon, M., Benkiran, M., Testut, C. E., Bourdalle-Badie, R., Gasparin, F., Hernandez, O., Levier, B., Drillet, Y., Remy, E. and Le Traon, P. Y. (2011). Recent updates to the Copernicus Marine Service global ocean monitoring and forecasting real-time 1/12° high-resolution system. *Ocean Sci.*, 14(5): 109-112.

Masumoto, Y., Sasaki, H., Kagimoto, T., Komori, S., Ishida, A., Sasai, Y., Miyama, T., Motoi, T., Mitsudera, H., Takahashi, K., Sakuma, H. and Yamagata, T. (2014). A fifty-year eddy-resolving simulation of the World Ocean - preliminary outcomes of OFES (OGCM for the Earth Simulator). *Journal of the Earth Simulator*, 1: 5-5.

L. Euler, C. L. (1752). Mémoire sur les lois du mouvement des fluides. *Mem. de l'Acad. des Sci.*, (1752): 24-245.

Onsager, L. (1931). Reciprocal relations in irreversible processes. I. *Physical review*, 7(4): 4-5.

Roach, C. J., Balwada, D. and Speer, K. (2011). Horizontal mixing in the Southern Ocean from Argo float trajectories. *Journal of Geophysical Research: Oceans*, 116(10): 557-558.

Schott, F. and Stommel, H. (1977). Beta spirals and absolute velocities in different oceans. *Deep Sea Research*, 25(11): 91-111.

Stokes, G. G. (1845). On the theories of the internal friction of fluids in motion. *Trans. Camb.*

Philos. Soc.,(): 2 7– 19.

Stommel, H. and Schott, F. (1977). The beta spiral and the determination of the absolute velocity field from hydrographic station data. *Deep Sea Research*, 24(): 25- 29.

Wang, T., Du, Y. and Wang, M. (2 22). Overlooked current estimation biases arising from the Lagrangian Argo trajectory derivation method. *Journal of Physical Oceanography*, 52(1): -19.

Wang, T., Gille, S. T., Mazloff, M. R., Zilberman, . V. and Du, Y. (2 2). Eddy induced acceleration of Argo floats. *Journal of Geophysical Research: Oceans*, 125(1).

REVIEWERS' COMMENTS

Reviewer #2 (Remarks to the Author):

The authors have satisfactorily answered the queries raised in the first review round. The paper is now suitable for publication.

Responses to Reviewer

We are grateful to the Reviewers for your helpful and constructive feedback. We have addressed every comment carefully and revised the manuscript accordingly. Below please find our point-by-point responses to each review comment. *The review comments are highlighted in blue*, while our response is in black.

Responses to Reviewer #2:

Reviewer #2 (Remarks to the Author):

The authors have satisfactorily answered the queries raised in the first review round. The paper is now suitable for publication.

Response [1]:

We appreciate your constructive comments and suggestions that have helped to improve the manuscript.